# AI-Guided Computing Insights into a Thermostat Monitoring Neonatal Intensive Care Unit (NICU)

**DOI:** 10.3390/s23094492

**Published:** 2023-05-05

**Authors:** Ning Zhang, Olivia Wood, Zhiyin Yang, Jianfei Xie

**Affiliations:** 1Faculty of Arts and Sciences, Beijing Normal University at Zhuhai, Zhuhai 519087, China; ningzhang@bnu.edu.cn; 2Galliford Try, Staffordshire Technology Park, Stafford ST18 0GP, UK; olivia.wood3@icloud.com; 3School of Computing and Engineering, University of Derby, Derby DE22 3AW, UK; z.yang@derby.ac.uk

**Keywords:** neonatal intensive care unit (NICU), thermostat sensor, computational fluid dynamics (CFD), turbulence modeling, artificial intelligence (AI), machine learning (ML)

## Abstract

In any healthcare setting, it is important to monitor and control airflow and ventilation with a thermostat. Computational fluid dynamics (CFD) simulations can be carried out to investigate the airflow and heat transfer taking place inside a neonatal intensive care unit (NICU). In this present study, the NICU is modeled based on the realistic dimensions of a single-patient room in compliance with the appropriate square footage allocated per incubator. The physics of flow in NICU is predicted based on the Navier–Stokes conservation equations for an incompressible flow, according to suitable thermophysical characteristics of the climate. The results show sensible flow structures and heat transfer as expected from any indoor climate with this configuration. Furthermore, machine learning (ML) in an artificial intelligence (AI) model has been adopted to take the important geometric parameter values as input from our CFD settings. The model provides accurate predictions of the thermal performance (i.e., temperature evaluation) associated with that design in real time. Besides the geometric parameters, there are three thermophysical variables of interest: the mass flow rate (i.e., inlet velocity), the heat flux of the radiator (i.e., heat source), and the temperature gradient caused by the convection. These thermophysical variables have significantly recovered the physics of convective flows and enhanced the heat transfer throughout the incubator. Importantly, the AI model is not only trained to improve the turbulence modeling but also to capture the large temperature gradient occurring between the infant and surrounding air. These physics-informed (Pi) computing insights make the AI model more general by reproducing the flow of fluid and heat transfer with high levels of numerical accuracy. It can be concluded that AI can aid in dealing with large datasets such as those produced in NICU, and in turn, ML can identify patterns in data and help with the sensor readings in health care.

## 1. Introduction

There are many reasons that a patient may need to be incubated, including premature birth, breathing difficulties, infection, diabetes, jaundice, long or traumatic delivery, low weight, or recovery from surgery. Preventing heat loss in newborn babies is critical for their survival. Heat loss occurs when blood flows closer to the surface, vasodilation occurs, sweat glands secrete fluid, and heat is lost to the environment via conduction, convection, radiation, or sweat evaporation. An incubator unit provides a safe controlled space for an infant to thrive whilst their vital organs develop; the unit maintains optimum temperature for the baby amongst other variables such as oxygen, humidity, and light [1].

The schematic in Figure 1 shows how a closed-system incubator works [2]. A fan blows the filtered ambient air over the heating element and the humidifier. Without the fan, heat cannot be conducted away from the heating element, causing the incubator to overheat. This moist, hot air flows into the plexiglass cabinet containing the baby, where the air escapes through the port holes and is processed again by the fan so that the process can start again. There is a 100–300 W thermostat-controlled heating element composed of a coiled resistance wire such as the kind used in autoclaves. The incubator heater has much less power than those used in autoclaves, so they do not get as hot. The thermostat consists of a thin capillary tube sensor and an expansion chamber that has a diaphragm. The sensor will not function properly if the capillary tube is rolled up and is not positioned this way. Usually, a liquid or gas is used in this system. When the liquid or gas is warmed up, expansion occurs, resulting in a rise of the diaphragm to activate a connected electrical switch.

The advantage of being an electronically controlled system is that an extra temperature sensor can be used and is attached with tape on the infant’s skin to monitor its body temperature. Some of the safety features integrated into an incubator are alarms that alert staff when the monitored systems deviate from the acceptable range. Overheating (above 40 °C), fan malfunction, and power failure set off these alarms to protect the baby. When using the incubator, it must be preheated for 30 min for climatization.

Computational fluid dynamics (CFD) is a useful tool for predicting flow and heat transfer in any form of intensive care unit (ICU). Beauchene et al. investigated the accumulation and transport of microbial-sized particles in a pressure-protected model burn unit [3]. Certain parameters need to be controlled within a permissible range in a burn’s ICU, such as pollutant concentration and air velocity, which should not exceed 0.2 m/s to prevent excessive drying at wound sites in an operating theater. Thermal convection flows [4], due to the temperature difference between the ICU and neighboring rooms, can transport airborne contaminants, creating a contaminated zone towards the ceiling. CFD can also be used to predict the contaminant distribution [5,6,7,8,9], and the predictions can help to implement control measures to reduce such contamination. Plexiglass curtains are used to channel the inlet air downwards into the room from the ceiling. The room has thermal sources of 70 W from the patient’s body heat, 135 W for each of the eight staff members, and 200 W for both ceiling lamps. Based on their research, room temperatures and ventilation systems were adjusted to reduce the risk of infection. Verma and Sinha also studied contaminant control in multi-patient ICU using CFD [10,11,12]. They pointed out that it was important to protect healthcare workers and patients against pathogens and infections.

Machine learning (ML) is applied artificial intelligence (AI) that provides systems with the ability to automatically learn and adapt from experiences without being explicitly programmed. This process concentrates on the development of computer programs that can access data and use it to learn for themselves. AI technology has many applications, such as weather forecast predictions using the sliding window algorithm [13] and for active climate control to improve the performance of heating, ventilation, and air conditioning units (HVAC) [14,15,16,17,18,19,20].

ML can be used to process large quantities of data in several fields. The fundamental laws can be expressed mathematically in the form of partial differential equations (PDEs), most of which cannot be solved analytically but can be solved by computing solvers numerically to optimize complex engineering solutions. CFD simulations allocate space using a grid to generate a mesh. As the grid density increases, so does the accuracy. It is important to consider the computational cost when increasing the accuracy of the solution: a balance should be maintained for simulations to remain efficient [21]. A data-driven approach can be paired with molecular simulations as a relatively new and effective way to derive governing equations in fluid mechanics [22], which means that data-driven analysis is becoming more important and should be researched. Furthermore, surrogate models have been widely used in CFD simulations. For example, surrogate model-based ML methods have been utilized for parameter estimation of the left ventricular myocardium [23] and for predicting flow parameters in a ventilated room using sensor readings [24].

Research into incubator technology and infant needs has suited the needs of patients admitted to the NICU. ML and automation are becoming increasingly popular in engineering and other fields, and this technique can be used in all intensive care units, not only inclusive to neonatal patients. ML can be used to learn patterns in temperature fluctuations in patients and feed this information into other systems such as a thermostat. Temperature control is fundamental to thermal comfort monitoring by a sensor of a thermoset [25]. CFD can be used to increase comfort levels and improve ventilation design. The combination of ventilation and high-quality air can inhibit infection, minimizing the spread of respiratory infections and general infections that patients have. The aim of this present study is to utilize the CFD data to train an AI model via ML strategies providing a better outlook on aerosol contamination dispersion characteristics to optimize the airflow patterns in NICU. Such models will tend to account for the dispersal mechanisms and focus on respiratory transmission in the next generation of hospitalization.

## 2. Three-Dimensional Modeling and Numerical Experiments

### 2.1. Three-Dimensional Modeling of NICU

#### 2.1.1. Climate Monitoring in NICU

The air needed in a NICU must be fresh and continuously circulating to avoid stagnant layer built up to keep room occupants (parents or neonates) at optimum comfort levels. The climate will be robust, and efficient heat exchange will be incorporated. Heat exchange is one of the most important aspects to consider in this study since neonates in need of incubation cannot regulate their body temperature by themselves due to being undeveloped and commonly prematurely born [26].

There are two major kinds of incubators: open and closed. The one modeled in this study is closed, and a schematic drawing of a closed construction incubator is shown in Figure 2. Hot air is blown around the baby through a canopy to keep the baby warm, and the temperature can be controlled using the external knobs or automatically via sensors on the baby’s skin. Homeostasis is the tendency to resist change to reach an equilibrium of the body’s internal environment via negative feedback. Thermoregulation is one of the major processes to be maintained for organs to work properly and controls the balance between thermogenesis (production of heat) and thermolysis (heat loss).

#### 2.1.2. Geometries of NICU and Incubator

Certain standards need to be adhered to in a NICU to provide the baby with the safest environment as much as possible. Regulations state that there must be 165 ft2 of floor space allocated per single-baby room [27]. Rooms can have multiple incubator units; this is dependent on how sick the baby is. This investigation is based on a single-baby room for a baby suffering from a more serious condition hence the single-baby style environment.

The room size, according to the health standards [27], must be at least 15.329 m^2^ for a single-baby room. The size of the computational domain is 16 m^2^, which satisfies this criterion and is large enough so that the flow patterns are not disrupted. The 3D model (see Figure 2) is based on real incubator dimensions and design inspired by Isolette^®^ 8000 plus Neonatal Closed Care Unit [28]. The dimensions of a simplified NICU, including an incubator and a radiator, can be found in Table 1, and the incubator is positioned in the center of the domain.

### 2.2. CFD Procedure

#### 2.2.1. CFD Simulations of Hospital Rooms

Kermani simulated the airflows and temperature distribution of a hospital room, focusing on maximizing the thermal comfort for all occupants of the room and minimizing the risk of airborne infection through indoor ventilation of good quality air [25]. The computational domain includes various furnishings, such as a wardrobe, lamp, bed, medical equipment (100 W/m^2^), inlet, and exhaust. A simplified patient on the bed and a doctor were modeled for other heat sources, and both had a constant heat flux of 60 W/m^2^. Kermani used an average temperature of 21 °C and a ventilation rate of 6 ACH, adhering to ASHRAE standard 170; both forced and natural ventilation were considered in their simulation. The inlet air temperature was 20 °C and exited the room through a ceiling-mounted grill acting as a pressure outlet. Efficient ventilation does not only increase comfort for residence but also can help to reduce the heating and cooling energy consumption of buildings. A quick increase or decrease in the temperature in a hospital room is also important.

Zhao et al. proposed an “N-point air supply opening model”, which is a simplified system based on a new air supply opening model (ASOM) and a numerical method of solving discrete algebraic equations to speed up and simplify the convergence procedure when predicting the airflows in a ventilated space [29]. They used the ASOM to describe the air inlet boundary conditions. Depending on the inlet air velocity, a variety of turbulent models can be used, i.e., a realizable *κ*-*ε* turbulence model. The solver process was very time consuming, but a multi-grid method had been developed to accelerate convergence for solving the non-linear algebraic equations. The proposed ASOM also has sub-divisions of models describing the boundary conditions in the vicinity of supply inlet. N-point ASOM has positives for both the direct ASOM and the momentum method.

#### 2.2.2. CFD Settings for NICU

The air conditioning unit used in this present study is based on Toshiba RAS-B07J2KVG-E indoor unit with an air mass flow rate of 0.1736 kg/s and an inlet velocity of 1.0 m/s [30]. The maximum operating temperature of the radiator used in the room is 120 °C, but it is set to a temperature of 50 °C, according to the common radiator settings to provide thermal comfort. The Reynolds number (Re) of the airflow is larger than 4000; therefore, it is treated as turbulent flow. The Mach number (Ma) is much less than 0.3, meaning that the flow is deemed incompressible. The values of the thermophysical parameters can be found in Table 2, and the boundary conditions are presented in Table 3. Commercial CFD solver Simcenter Star-CCM+ is used to solve the governing equations via an integral discretization method, i.e., the finite volume method (FVM) [31,32], and Table 4 lists the physical models used in our CFD simulations. Besides the pre-processor, we have specified several 1D line probes and 2D plane sections to visualize the convective flows inside a NICU while post-processing the data. As shown in Figure 3a, four vertical lines (a–d) and four horizontal lines (e–h) have been derived to indicate the exchange of momentum and heat transfer at different locations.

### 2.3. AI Model Development

#### 2.3.1. AI for CFD Simulations

Deep learning (DL) is a bifurcation of ML where artificial neural networks (ANN), algorithms inspired by the human brain, learn from large amounts of data. DL can be used to discover the nonlinear partial differential equations (non-PDEs) from scattered and potentially noisy containing random or irrelevant data observations in space and time [33]. Complex fluid dynamics and heat transfer problems depend on numerical simulations such as CFD, i.e., by spatially and temporally discretizing the governing equations (Navier–Stokes–Fourier equations) and reducing a system of PDEs to an algebraic system. However, CFD engineers have to pay computational penalties in real-life applications, such as climate change in hospital rooms and design optimization in manufacturing process, due to complicated geometries and custom demands. DL has shown new promises for surrogate modeling due to its capability of handling strong nonlinearity and high dimensionality. Sun et al. provide a physics-constrained DL approach for surrogate modeling of fluid flows without relying on any simulation data via structuring a deep neural network (DNN) architecture [34], which is devised to enforce the initial and boundary conditions. They incorporated the governing PDEs into the loss of the DNN to drive the training. They reported excellent agreement on the flow field and forward-propagated uncertainties between the DL surrogate approximations and the first-principle numerical simulations.

ML can be used to aid decision making by neonatologists in a NICU. Data is constantly streamed and analyzed for the right judgment of the patient to be made, in unison with the medical team’s critical thinking, for the baby to be discharged as soon as possible. There are four ways that ML is being applied within hospitals. The first is predicting birth asphyxia, which is when the infant receives a lack of oxygen. Speech recognition techniques are used to detect early signs of birth asphyxia in a newborn baby’s cry. A prototype (with 89% accuracy) has been developed to support the vector-machine-based ML model that can correctly classify the recordings of known asphyxiating infants. Other areas of prediction using ML include detection of seizures, sepsis, and the prediction of respiratory distress syndrome.

It is great of interest for CFD engineers to apply an AI model to accelerate their simulations. More recently, Monolith AI, the leading platform for design and engineering, and Simcenter STAR-CCM+, provided by Siemens, have joined forces to develop advanced computing capacities for CFD engineers with applications in different fields, such as aerospace and automotive engineering. In this study, we consider a proof of concept with Monolith AI to further what is possible in a design exploration study on a thermostat sensor to monitor climate change in a NICU by training an AI model in terms of ML strategies from the datasets obtained in Simcenter STAR-CCM+ simulations.

#### 2.3.2. ML in an AI Model

There are many challenges that CFD engineers face while monitoring climate change efficiently in a NICU: a variety of geometric parameters, such as dimensions of NICU and incubator, and varying thermophysical variables in convective flows. These concerns on the geometry of the domain and different elements and physics of the flow itself really stretch the widely used commercial CFD solver such as Simcenter Star-CCM+. The advantages of an AI model or platform such as Monolith AI enable us to sufficiently dismiss the challenges. Therefore, the solution to these problems is to create an AI model which can take important geometric parameter values as input from a CFD configuration, i.e., dimensions of NICU and incubator. Then, we can train the AI model in ML in terms of the data obtained in our CFD simulations. As a result, more accurate predictions of climate change (i.e., the temperature evaluation), which take less time compared with the convectional CFD techniques, will be provided by the developed AI model in real environment of hospital rooms such as NICU.

Furthermore, this present work not only concentrates on the prediction of a single scalar (e.g., the pressure) or vector (e.g., the velocity) for a room but also focuses on the evolution of results over time for the full cycle of climate change in a NICU. It needs to make an effort to plan both our CFD simulation and AI modeling, which will showcase the capacity to monitor the climate more thoroughly in real time.

Among the thermophysical properties in this study, which describe the evolution of airflow and heat transfer, there are three parameters of interest to be selected: the mass flow rate (i.e., inlet velocity), the heat flux of radiator (i.e., heat source), and the temperature gradient caused via the convection. These parameters will moderate the uniformity of the introduced flow and strength of the heat transfer throughout the incubator. To do so, we need to divide the data from CFD simulations into two parts: one is for training, and the other is for validation. Moreover, the AI model will be stretched further to make more general predictions, i.e., generality of an AI model. For instance, we train the AI model not only in the most turbulent flow cases but also to validate the ML’s expected data in cases where a sharp temperature gradient exists. This training procedure helps challenge the AI model to make it more general to predict the physics of fluids such as a convective airflow in a NICU.

There are multiple ML models available in the Monolith platform (Artificial Neural Networks, Random Forests, and Polynomial Regressions, etc.), and the selection depends on many things, i.e., the data format, the data quantity, and the complexity of the problem. We chose the Artificial Neural Network (ANN) as our AI model in the ML method and tested it in this present work. The workflow and flow of data between a computational tool (i.e., the CFD solver) and AI model (i.e., Monolith AI) are given as follows:We first run CFD simulations using Simcenter STAR-CCM+ (2022.2) to collect a group of data regarding the geometric parameters and boundary conditions in a NICU, plus a reliable turbulence model (i.e., the *κ*-𝜀 model). The initial samples from CFD simulations are presented on the left panel of Figure 4;Then, we extract the necessary data from CFD simulations and rearrange the dataset (i.e., inputting a tabular notebook);Next, Monolith AI is fed by the prepared data as inputs to develop an AI model, i.e., data training in ML via adopting an ANN model. The training process and data exporting are presented on the top right panel of Figure 4;At last, the newly developed AI model is utilized to forecast climate change in a NICU with a high level of numerical accuracy, comparable with the traditional CFD tools but less time consuming (i.e., AI accelerated CFD);Figure 4 also indicates how we can benefit from the active control of a comfort NICU via a sensor reading contributed by ML on an AI platform. This is illustrated in the bottom right panel of Figure 4.

#### 2.3.3. ANN features in ML with Monolith AI

In the architecture of ANN [35,36], the data are passed through a series of connected neurons, and the model can learn to perform tasks or find relationships crossing the data. With a large enough dataset, ANN can learn to predict extremely complex non-linear relationships such as the non-PDEs in fluid dynamics and heat transfer problems in a NICU. It is noted that ANN has many parameter choices that can affect the accuracy and performance of the model predictions. However, Monolith platform has customized these choices and narrowed them down to the most common ones for their users. As with ANN provided by Monolith, we first select a dataset to train our model. Then, we choose the input and output columns for the model to learn, as shown in Figure 4.

Usually, ANN learns to perform tasks from considering examples, generally without being programmed with task-specific rules (i.e., code-free ML services). The architecture (see Figure 5) is normally composed of many layers consisting of many neurons. One of the most important choices is the shape of the network. The neurons in an ANN are arranged in a series of layers known as “hidden layers”. In the Monolith platform, we choose both the number and the size of these hidden layers. Monolith helps us train and compare many models with different architectures, ensuring that the selected one gives us the best performance both on training data and on unseen testing data. The second most important choice is how many training steps to use. Using more training steps will make the model fit more closely to the training data. It is usually expected to make the model predictions match the training data more closely.

ANN looks at the data in chunks or batches. The number of rows in each batch (i.e., batch size) can affect how quickly the model trains and also the final performance. We use a common default size of 32 rows per batch, but it allows us to use larger or smaller values to see how this affects the model performance. It is suggested to take the values between 16 and the full dataset size. It should be noted that one may see much slower training times if the batch size is larger than one thousand.

During the training phase, the different weights between these neurons will be progressively tuned so that the network is able to accurately predict the outputs (i.e., AI model) from the inputs (i.e., the CFD samples). In general, if the structure is too small, the problem will be simplified, which might lead to inaccurate predictions of data. On the opposite, if the structure is too large and too complex, the network will “overfit” to the current data, which might lead to errors in new data. All the CFD simulations and AI training have been performed upon an average workstation, i.e., Lenovo ThinkPad P15 Gen 1 Mobile Workstation: Intel (R) Core (TM) i7-10850H vPro Processor (12MB Cache, up to 5.10GHz, 6 Cores), 16.0GB Installed RAM, and 256GB Solid State Drive.

## 3. Results

In the post-processing of the thermophysical properties, our CFD results are focused on the region only inside the incubator, where the thermostat sensor is more sensitive to the climate change. To do so, a vertical plane centrally crossing the incubator is created to effectively display the airflows and temperature evolution in the incubator, as shown in Figure 3b, which can be analyzed to understand the results in further detail, and its corresponding dimensions are listed in Table 1. Furthermore, a line probe that is 35 mm above the infant is created to demonstrate the performance of the turbulence modeling in terms of a realizable *κ-*𝜀 model in RANS, such as the turbulent kinetic energy (TKE) and turbulent viscosity.

### 3.1. Velocity Field

The mass flow rate of the air is important to newborn babies and is essential to their health care. Therefore, it is of great interest to predict the air conditions, such as flow speed in an incubator. As seen in Figure 6b, more uniform streamlines are recovered above the cylinder (i.e., a modeled infant), and the structures of the vortex are visualized more efficiently. Importantly, based on the data from ML in an AI model, the maximum speed of the air passing through the incubator has been improved to a more accurate value of 0.164 m/s (e.g., Figure 6b) compared with one of 0.146 m/s obtained in CFD simulations (e.g., Figure 6a).

### 3.2. Pressure Contour

Due to the exchange of momentum between the air and incubator, the flow has been slowed down on the left side wall of the incubator. Accordingly, a large pressure contour is observed in front of the incubator (i.e., the left view.). As seen in Figure 7a, the maximum pressure exits in front of the left inlet of the incubator. A similar contour of the pressure is also observed in Figure 7b, but the maximum value of pressure has been predicted to be 0.0138, which is larger than the value of 0.0125 obtained in CFD simulations. The data of pressure from ML returns a more accurate value of the pressure compared with the CFD results.

### 3.3. Temperature Distribution

Apart from the mass conservation of air and momentum exchange between the air and incubator, the temperature evolution occurring in a heat transfer process such as the convection hints to us whether a comfortable environment exists in a NICU, particularly a healthy recovering condition for a newborn baby or patient. As discussed previously, the results in this present work are only focused on the regions in which the incubate is located, as the temperature of other regions in a NICU would not be changing sustainably. This is indeed the case since a uniform distribution of the air temperature surrounding the incubator is shown in Figure 8. Based on the temperature evolution indicated in Figure 8, it is the same situation in front of the cylinder (i.e., the modeled infant) for both the CFD results and data from ML. However, the heat transfer has been enhanced significantly at the foot region of the modeled infant, which is illustrated in Figure 8b. The enhancement of the heat transfer in convection can help alarm the sensor of the thermostat at an early stage to take away the extra heat flux produced by the infant if they have high temperatures.

### 3.4. Turbulent Kinetic Energy (TKE)

The flow inside and around the incubator is complicated and turbulent. The turbulent kinetic energy (TKE), measuring the intensity of turbulence, along a line probe sitting 35 mm above the modeled infant is shown in Figure 9. As seen in Figure 9, TKE is invariant for both the CFD results and data from ML at the first quarter of the line probe, i.e., *x* = −0.4, before the air reaches the modeled infant. However, the TKE increases significantly according to both the CFD simulations and ML predictions. Furthermore, the data in ML show lower TKE initially but larger TKE after the central location of the modeled infant compared with the CFD results.

### 3.5. Turbulent Viscosity

Following from the TKE plot in Figure 9, we have further investigated the turbulent viscosity along the same line probe centrally crossing the incubator. As shown in Figure 10, there is no big difference in the turbulent viscosity between the CFD results and data predicted in ML before the central point of the probe line. However, the values of turbulent viscosity given by ML started increasing earlier compared with the CFD results. Nevertheless, the magnitude of turbulent viscosity obtained from both CFD results and ML data is close to each other, and the development of the CFD dataset is 0.3 m behind that of ML, which is about 17% of the length of the incubator.

## 4. Discussion

Many things affect heat and mass transfer in energy balance models, including layers of clothing worn, metabolic rate, air temperature, mean radiant temperature, air speed, and relative humidity. Thermal comfort is a state of mind, however, which is unique to each person; it does not relate to the heat, mass transfer, and energy balance equations. In a hospital environment, individuals are more susceptible to infection due to the proximity of other patients, hence the airborne transmission of their infection.

The present simulations have included different heat sources: radiator and human. This present study suggests that an AI model can accurately predict the air velocity and temperature distribution in most regions of a simplified NICU. It has also demonstrated that CFD can be used to study airflow and heat transfer in isolation rooms for highly contagious patients in hospitals, ensuring that negative pressure is maintained in isolation rooms to contain contaminants inside this room to prevent the spread to non-isolated patients. Besides the airflow and temperature evolution, the turbulence intensity has also been investigated via plotting TKE and the turbulent viscosity.

Mesh resolution (fine mesh) is needed to guarantee good numerical accuracy in conventional CFD simulations. However, simulation with a fine mesh is time-consuming, and how to refine the mesh locally to improve the accuracy and reduce the computing time is always challenging for CFD engineers (see Figure 11a), particularly when the geometries are complex and have many curved boundaries. Nowadays, commercial CFD solvers provide their users with good tools to generate the mesh, but it still requires training on how to deal with complicated geometries, which may take years to become experienced CFD engineers. Apart from the discretization concerning mesh generation, it is important to recover the physics in fluid flow and heat transfer more efficiently, such as the selection of a reliable turbulence model. AI has great potential in both fluid dynamics and heat transfer communities to avoid paying the computational penalty in massive engineering computing such as climate change in a NICU. For CFD engineers, there is a great benefit from AI technologies to accelerate our computation in terms of better usage of existing data produced in CFD simulations, as shown in Figure 11b.

It will augment the productive outcomes if an AI model is integrated into a CFD application. In this present study, our CFD simulation data has been used to train an ML model to make real-time predictions of climate change in a NICU, including the airflow and temperature evolution. Based on the trained AI model, we have made comparisons of the monitoring variables between the ML outputs and original data produced via CFD simulations. To do so, we have restricted a group of data, starting with the AI model training and then revisiting it until the training process is finished. The trained AI model is fed the inputs from the CFD case studies and makes its own outcomes on what the output responses are. This is how an AI model has been used to accelerate simulations of fluids with static geometry (i.e., dimensions of NICU, incubator, and radiator) and varying boundary conditions (i.e., the mass flow rate, perfectly insulant walls, and the heat flux). The scale of speedup via an AI model is presented in Table 5 compared with the traditional CFD solver, and the speeding ratio is comparable to a reported value of 8.0 [37]. The last step of the procedure is to compare these outputs with the CFD results, which are used as physics-informed (Pi) baselines in our validation.

As shown in Figure 6, Figure 7, Figure 8, Figure 9 and Figure 10, the CFD results from Simcenter STAR-CCM+ and the ML predictions are plotted together for climate change in a NICU from the training set. The velocity, pressure, temperature, TKE, and turbulent viscosity are predicted. The results are almost indistinguishable in some areas between the CFD simulations and AI outputs. Furthermore, the AI model predictions make a positive comparison with the ground truth data (i.e., the CFD results). It can be, therefore, drawn that after data training is completed in AI, predictions from ML can be made on new cases in real time.

One of the limitations is the selection of the AI models, such as the ANN method in our AI training procedure, and it should test other AI models (i.e., Random Forests and Polynomial Regressions) in the ML approach to demonstrate their advantages and disadvantages over the accuracy and uncertainty analysis to create effective test plans informed from the historical data, raw data, training data, structured data or even unstructured data and to predict when enough data has been captured for results to be gathered from the code-free AI platforms such as Monolith AI. Future work will include the product design optimization of thermostats in a NICU in terms of the more accurate thermophysical properties such as air the mass flow rate and temperature change in AI-speedy CFD simulations. Furthermore, regarding the physics of flow in any hospital room, it is worth noting that AI can help avoid running many CFD simulations of fluids with static geometries and varying boundary conditions to efficiently detect if sensors are not mounted or connected properly for data preparation (i.e., Internet of Things) and to quickly identify and correct drift in measurements with our AI models (i.e., zero carbon footprint). This helps assess the sensitivity of a design with respect to its geometric parameters or boundary conditions to understand how robust it will be from a manufacturing or operational standpoint.

## 5. Conclusions

Many indoor simulations for airflow and heat transfer had been carried out previously, but the climates inside a NICU had not been extensively studied with a CFD approach. This present study has modeled and predicted the climate of a three-dimensional (3D) NICU in terms of CFD strategies, such as a realizable *κ*-𝜀 turbulence model, which is suitable for indoor climate simulations and is widely used by the CFD community.

This present study has investigated the flow patterns and temperature distributions in a NICU. Ambient air temperature, heat sources, and inlet air velocity are all factors that affect the flow structures of an environment. As the heat from the radiator is transferred to the ambient air along with the inlet air, heat transfer takes place and the room temperature starts to increase The thermal equilibrium has been eventually reached until there is no longer an increase in room temperature due to the heat source.

In terms of the data-driven strategies in an AI model for CFD, neonatologists in a NICU can utilize ML to aid the critical decision making on the ward. Data is constantly streamed and analyzed for the best assessment of infants, ensuring they are discharged as soon as safely possible. ML has been used to predict birth asphyxia, detect seizures, sepsis, and respiratory distress syndrome. Body sensors can be fixed to infants to track their body heat and breathing. AI is very helpful when dealing with large datasets such as those produced by CFD simulations in this environment, and in turn, ML can identify patterns in data and be used to make predictions in healthcare settings.

Empirical modeling provides correlation functions within rich data. AI and ML are excellent at empirical modeling. This work combines the physical principles and empirical modeling into a unified approach: Pi data-driven methods for multi-physics optimization. The ML solutions will not violate the physical constraints. The proposed computational framework has been applied to reconstruct the high-resolution flow images (i.e., the ML expected data) from the low-resolution input (i.e., the CFD results).

In conclusion, the post-processing of this CFD application shows the sensible flow structures and heat transfer as expected by any indoor climate with this configuration. Based on the data-driven method in ML, i.e., the CFD results presented in Section 3 within the Monolith AI interface, an AI model can be utilized to augment the design workflows and cut the design period with the help of rich data available. The AI-guided computing insights have been implemented to demonstrate how AI and ML have accelerated the CFD application in a NICU. There is no doubt that ML and the Internet of Things (IoT) will not only enable to monitor the comfort zone of patients but also diagnose the disease in clinics and manage the thermal analysis in terms of the thermostat such as a smart sensor (i.e., wearable devices).

## Figures and Tables

**Figure 1 sensors-23-04492-f001:**
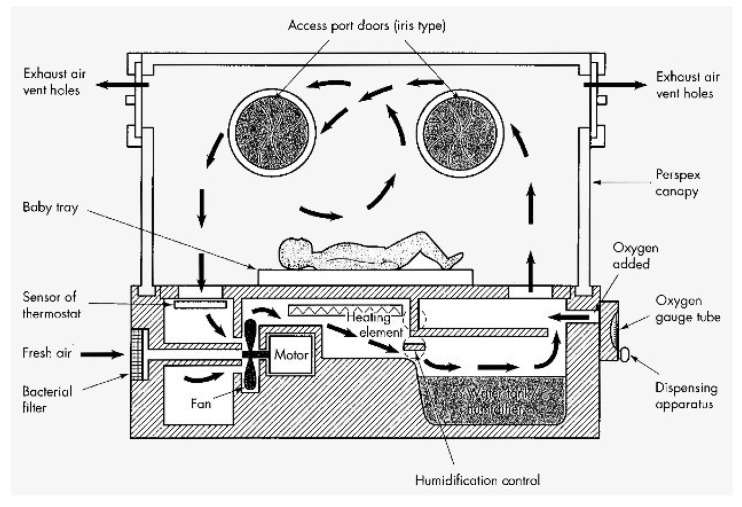
Typical schematic of an incubator integrated with the sensor of thermostat and humidification control [2].

**Figure 2 sensors-23-04492-f002:**
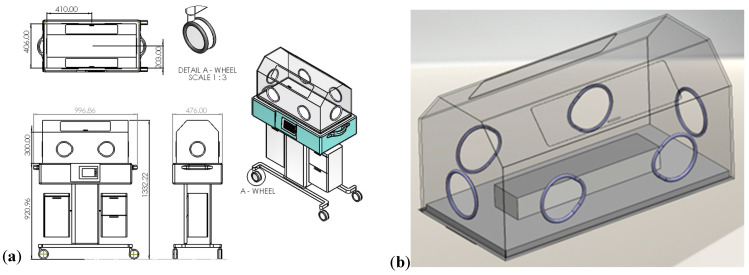
Three-dimensional model of a closed incubator in a NICU (**a**). (**b**) Zoom into the incubator: a cylinder (510 × 110 × 70) is used to simulate an infant laying on a bed. All dimensions are in mm.

**Figure 3 sensors-23-04492-f003:**
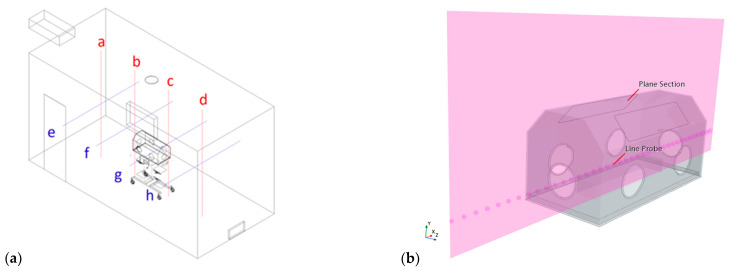
Locations of line probes (four verticals: a, b, c, and d; four horizontals: e, f, g, and h) across the NICU (**a**) and plane section (in purple) and line probe (dotted line) crossing the incubator (**b**)**.**

**Figure 4 sensors-23-04492-f004:**
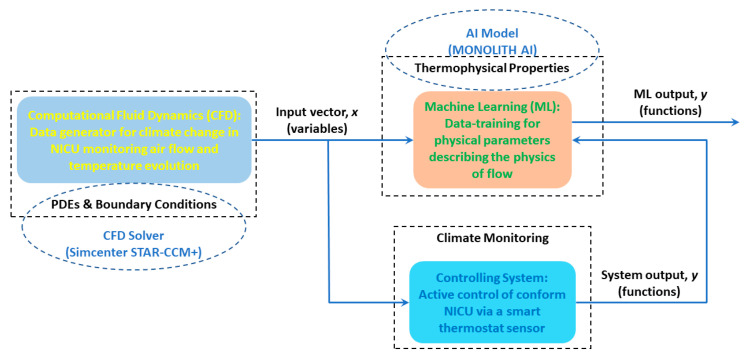
Physics-informed (Pi) ML in an AI model accelerating CFD simulations.

**Figure 5 sensors-23-04492-f005:**
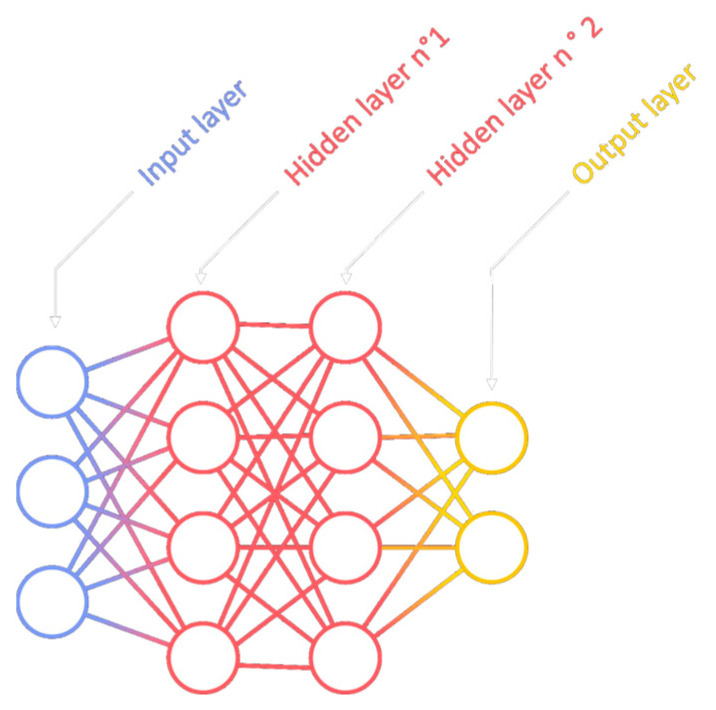
Schematic of artificial neural network (ANN). The different layers of an ANN are as follows: Input layer (in blue) gathers all the inputs that will be learned from (e.g., the speed of air and length of incubator); Hidden layers (in red) in which the neurons will be connected; Output layer (in yellow) contains the expected outputs (e.g., the pressure and temperature). Reproduced from Monolith AI platform.

**Figure 6 sensors-23-04492-f006:**
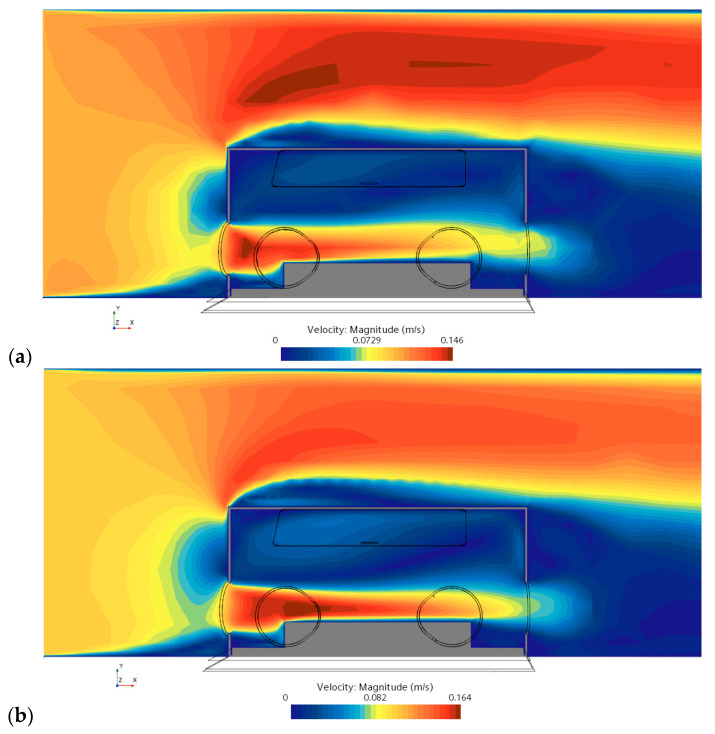
Airflow displayed in a vertical plane centrally crossing the incubator: (**a**) CFD results and (**b**) expected data from ML in an AI model.

**Figure 7 sensors-23-04492-f007:**
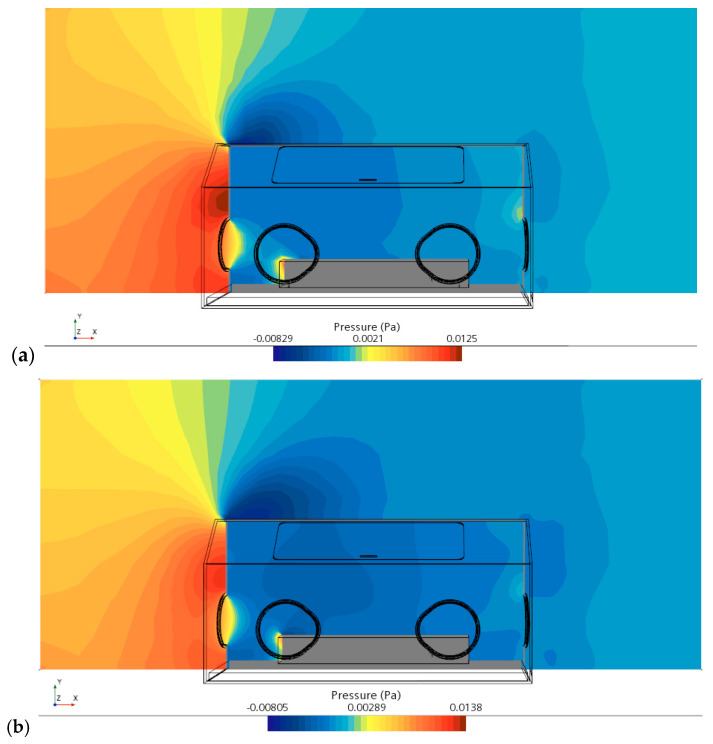
Pressure contour displayed in a vertical plane centrally crossing the incubator: (**a**) CFD results and (**b**) expected data from ML in an AI model.

**Figure 8 sensors-23-04492-f008:**
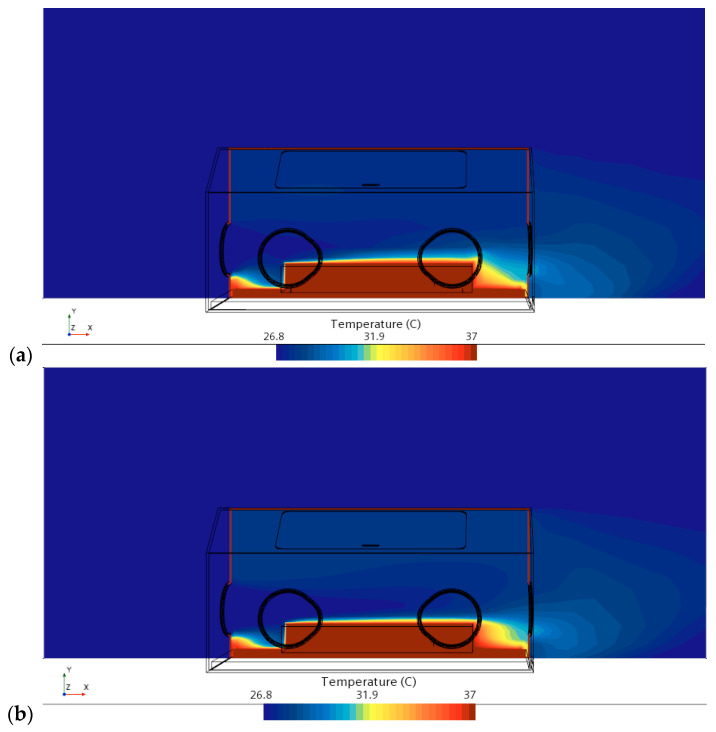
Temperature evolution displayed in a vertical plane centrally crossing the incubator: (**a**) CFD results and (**b**) expected data from ML in an AI model.

**Figure 9 sensors-23-04492-f009:**
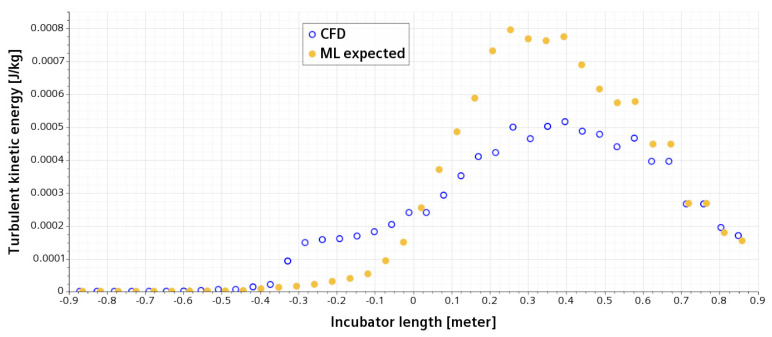
Variety of turbulent kinetic energy (TKE) on a horizontal line, which is 35 mm above the infant and centrally crossing the incubator; open circles for CFD, and filled circles for the expected data from ML in an AI model.

**Figure 10 sensors-23-04492-f010:**
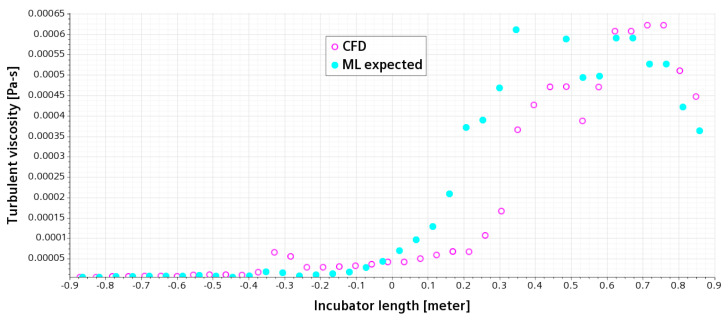
Variety of turbulent viscosity on a horizontal line, which is 35 mm above the infant and centrally crossing the incubator; open circles for CFD, and filled circles for the expected data from ML in an AI model.

**Figure 11 sensors-23-04492-f011:**
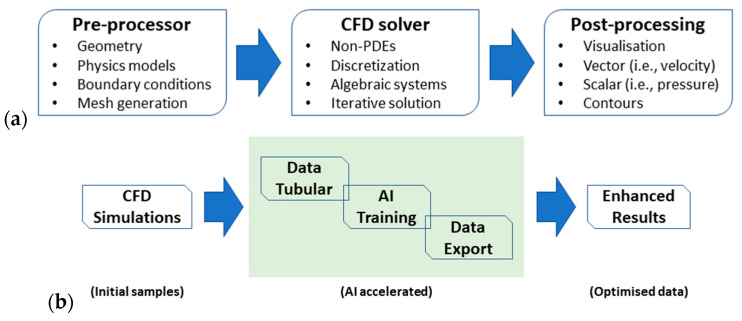
Scheme of conventional non-AI approach (**a**) versus AI-accelerated simulation (**b**).

**Table 1 sensors-23-04492-t001:** Geometric parameters of a simplified NICU, including incubator, inlet/outlet, and radiator.

Case	Length [m]	Width [m]	Height [m]
NICU	5.5	2.5	3.0
Incubator	0.997	0.476	1.332
Inlet/outlet	0.5	0.04	0.25
Radiator	0.9	0.12	0.6

**Table 2 sensors-23-04492-t002:** Thermophysical properties of the parameters used in this present study.

Parameter	Symbol	Value
Mass density	*ρ*	1.204 kg/m^3^
Specific heat	*C_p_*	1007 J/(kg K)
Thermal conductivity	*κ*	0.02514 W/(m K)
Thermal diffusivity	*α*	2.074 × 10^−5^ m^2^/s
Dynamics viscosity	*µ*	1.825 × 10^−5^ kg/(m K)
Thermal expansion coefficient	*β*	3430 K^−1^

**Table 3 sensors-23-04492-t003:** Boundary conditions of the computational domain and elements in NICU.

Boundary	Velocity/Pressure	Temperature
All side walls, ceiling, and ground floor	*u = v = w =* 0 ***	insulated
Velocity inlet	*u =* 1.0 m/s	20 °C
Pressure outlet	constant	-
Radiator	stationary	50 °C
Infant	stationary	37.5 °C

* *u*, *v*, and *w* are the velocity components along *x*, *y*, and *z* directions, respectively.

**Table 4 sensors-23-04492-t004:** Physics models selected for the airflow and convection in CFD simulations.

Category	Model
Space	three-dimensional (3D)
Time	steady state
Material	air
Equation of state	ideal gas
Turbulent model	realizable *κ*-𝜀 in Reynolds-Averaged Navier–Stokes (RANS)

**Table 5 sensors-23-04492-t005:** Comparison of computing times between the CFD solver only and an AI-accelerated model.

Module	CFD Solver Only	AI-Accelerated Model	Speedup
Accumulated CPU Time Over All Processes (s)	5282.23	880.34	6.0

## Data Availability

Not applicable.

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
