# Peer review of "AI-Guided Computing Insights into a Thermostat Monitoring Neonatal Intensive Care Unit (NICU)"

_sensors, 2023, doi:10.3390/s23094492_

Round 1
Reviewer 1 Report
The manuscript investigates the thermostat monitoring of NICU. CFD was used to simulate the airflow and heat transfer and a surrogate model was trained and validate. The manuscript can be improved from following aspects.
1. The surrogate modelling methods have been widely used in CFD. This should be discussed in the introduction.
2. The ML model structure should be provided. Otherwise, readers cannot reproduce the method.
3. Since authors do not provide their model, it is not clear whether the ML model can reproduce the dynamics of the NICU or just the steady state.
4. Figure 4 seems to show a model predictive control of the thermostat. However, this has not discussed in the text.
5. Authors claimed the surrogate model has reduced the computational cost. This should be compared and discussed.
6. Authors claimed the prediction is accurate. What were the criteria?
Reviewer 2 Report
The manuscript presents a comprehensive study on monitoring and controlling airflow and ventilation inside a neonatal intensive care unit (NICU). The use of computational fluid dynamics (CFD) simulations is appropriate for this study, and the incorporation of machine learning (ML) in an artificial intelligence (AI) model provides accurate predictions of the thermal performance associated with that design in real-time. The manuscript highlights the importance of the geometric parameters and thermophysical variables in capturing the physics of convective flows and enhancing heat transfer throughout the incubator. However, there are a few points that could be addressed to improve the manuscript.
Firstly, in the introduction section, the use of ML includes a broader area like Siri, Alexa or Tesla, which can be narrowed down to other NICU or even ICU use cases. In section 2.3.1 Deep learning was introduced abruptly, that paragraph should be rewritten.
Secondly, a more detailed description is required for the Data processing from the CFD simulations. This is unclear in the current state.
Thirdly, the authors should provide more details on the data used for training and validation of the AI model, including the size of the dataset and characteristics. Lastly, it would be beneficial to include some discussion on the limitations and potential future directions of the study.
Overall, this manuscript presents a valuable contribution to the field of healthcare and engineering, and I recommend its publication with revisions.
